# POCUS Diagnosis of Sternal Fractures in Children without Direct Trauma—A Case Series

**DOI:** 10.3390/children9111691

**Published:** 2022-11-03

**Authors:** David Troxler, Johannes Mayr

**Affiliations:** 1Pediatric Emergency Unit, University Children’s Hospital Basel, CH-4031 Basel, Switzerland; 2Pediatric Surgery Department, University Children’s Hospital Basel, CH-4031 Basel, Switzerland

**Keywords:** sternal fractures, POCUS, children, indirect trauma

## Abstract

Sternal fractures in children are rare and tend to be caused by high-energy injury. However, sternal fractures may also be caused by minor and indirect trauma and might therefore not be as rare as previously stated. Ultrasound examination is the method of choice for the diagnosis of sternal fractures. We present three cases of sternal fractures not caused by direct trauma that were presented to our A&E department within a period of only 62 days. All of them exhibited localized tenderness, but none had an associated injury. Children presenting to the A&E unit with musculoskeletal thorax pain should be screened by POCUS for sternal fractures, even if they do not report any direct trauma to the thorax.

## 1. Introduction

Sternal fractures in children are traditionally considered a rare event and are usually associated with high-energy injuries mainly caused by motor-vehicle incidents [1,2,3]. However, sternal fractures might also be caused by minor or indirect trauma. These injuries may not be visible on X-rays and are, therefore, often overlooked [4].

Ultrasound examination of the sternum is well-established and has proven to be more sensitive than X-ray or even CT for diagnostic imaging of sternal fractures [5,6,7,8]. With the progressive use of point-of-care ultrasound (POCUS), some case reports have emerged documenting sternal fractures after low-energy trauma or even in the absence of direct trauma to the sternum or thorax [9,10,11,12].

We present a series of three cases of sternal fractures that occurred without direct trauma to the thorax. The three cases were presented to our A&E department within a period of only 62 days.

## 2. Case Report

Case 1: A 14-year-old boy presented to the A&E unit after a jump with a somersault from a height of 2 m. Upon landing on his feet on an air mattress, he felt his chest “collapse”. Since then, the boy had complained of pain across the sternum that was exacerbated by arm movement and deep inspiration. He did not receive any direct impact to the chest. Physical examination revealed pain across the sternum, and POCUS showed a fracture of the caudal part of the manubrium sterni (Figure 1).

Case 2: A 9-year-old boy presented to the A&E department because of persistent pain across the sternum after falling from monkey bars and landing on his feet. He did not sustain any direct trauma to the thorax. He complained of pain, especially during physical effort and laughing. Upon clinical examination, he complained of pain across the sternum and the right sternocostal region. POCUS revealed a fracture of the cranial part of the first sternebra (Figure 2).

Case 3: A 6-year-old girl presented to the A&E department after attempting a somersault on the trampoline and landing on her head. She complained of pain across the anterior thorax. Upon physical examination, pain could be elicited across the manubrium sterni and the cranial part of the sternum. POCUS revealed a fracture of the caudal part of the first sternebra (Figure 3). The girl did not exhibit any signs of cerebral concussion.

None of the three patients exhibited any associated injuries. No skin changes or hematomas were seen in any of the patients. The palpation or percussion of the spine did not elicit any pain in all three children. All three patients received symptomatic pain treatment and were discharged from the A&E unit on the same day with a recommendation of sports restriction for 6 weeks.

For this article, we conducted a telephone follow-up with patients’ families 6 months after their initial presentation. No follow-up consultation was necessary, nor were any complications reported in any of the children presented. All had returned to full activity after the recommended restriction period.

Table 1 shows the baseline characteristics and mode of injury of the three children suffering sternal fractures.

## 3. Discussion

As sternal fractures in children may not only be caused by direct trauma but may also be induced by minor and indirect impact, they appear to be more frequent than previously stated. Hyperflexion of the cervical and thoracic spine with longitudinal compression seems to be a frequent mechanism of injury [13].

POCUS provides a fast and reliable diagnosis of sternal fractures without the use of ionizing radiation. Associated lesions seem to be rare in sternal fractures caused by minor, indirect deforming forces. These fractures can be managed safely in an outpatient setting.

A clear diagnosis by POCUS imaging benefits the patient because it allows the patient to adapt activities accordingly.

## 4. Conclusions

As POCUS is fast and devoid of inherent risks of ionizing radiation, A&E departments should have a low threshold in screening children with musculoskeletal thorax pain for sternal fractures.

Further research is needed to evaluate the incidence, risk of associated lesions, and optimal duration of restricted activity as well as the prevention of such injuries in high-risk activities.

## Figures and Tables

**Figure 1 children-09-01691-f001:**
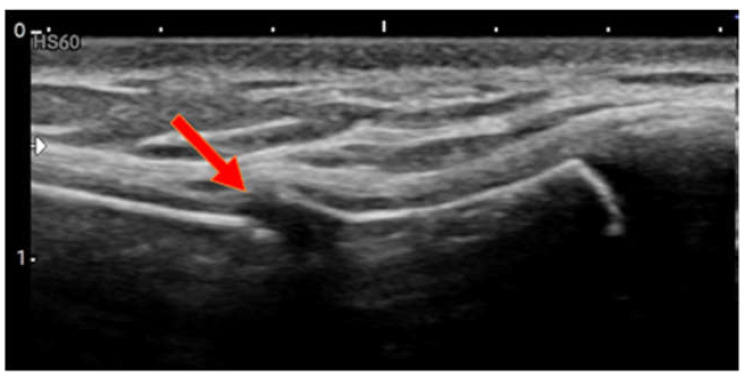
Case 1, fracture of caudal manubrium sterni.

**Figure 2 children-09-01691-f002:**
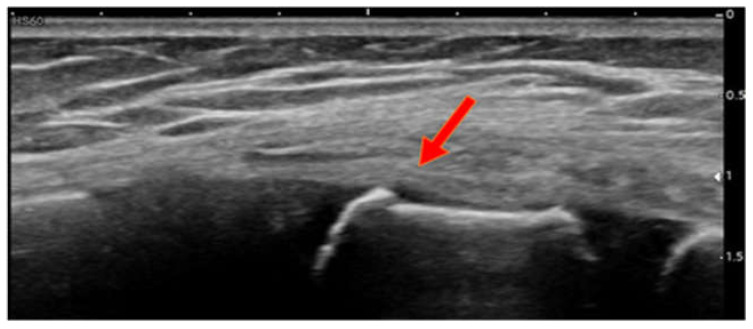
Case 2, fracture 1st sternebra cranial.

**Figure 3 children-09-01691-f003:**
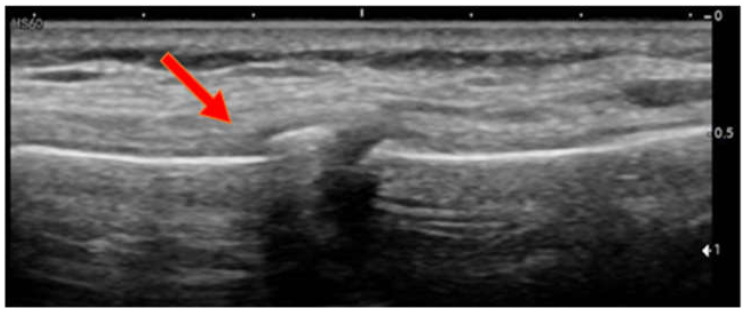
Case 3, fracture 1st sternebra caudal.

**Table 1 children-09-01691-t001:** Baseline characteristics and mode of injury of the three children suffering sternal fractures.

	Case 1	Case 2	Case 3
**Age (years)**	14	9	6
**Sex**	Male	Male	Female
**Trauma impact**	Feet	Feet	Head
**Fracture site**	Manubrium sterni caudal	1st sternebra cranial	1st sternebra caudal
**Displacement**	None	None	None

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
