# Peer review of "POCUS Diagnosis of Sternal Fractures in Children without Direct Trauma—A Case Series"

_children, 2022, doi:10.3390/children9111691_

Round 1
Reviewer 1 Report
This is an interesting case series, I just have a few questions for the authors:
Was the skin over the sternum normal? Did the present a hematoma?
How long was their follow -up?
Did the authors consider repeating the US? If so, when?
Author Response
Dear Reviewer
We thank you for your careful review of our article and for your insightful comments.
- The skin over the sternum did not show any hematoma or other changes in any of the patients.
- There was no follow-up planned at the beginning. For the purpose of the article, all patients were followed up by phone. None of them had experienced any complications or needed a return visit to the clinic or their pediatrician. All were back to full activity after the recommended rest time (4-6 weeks)
- We did not repeat any US examinations as we did not have any records of follow-up visits for those patients.
We added this information in the article.

Reviewer 2 Report
Dear Authors,
Thank you for this simple yet important manuscript. This work of yours is adequate to create awareness but inadequate with regards to what factors and findings you look for on follow up of the patients. What were the clinical findings and the appearance of the ultrasound on follow up of patients? I believe you are in A&E setting and you are highlighting POCUS as a tool. May I suggest you revise the title so that this is clearer to the readers that you are focusing on diagnostic tool. Case series are usually more detail and the loop is completed to include evidence of full recovery clinically and objectively
All the best!
kind regards,
Reviewer 2
Author Response
Dear Reviewer
We thank you for your careful review of our article and for your insightful comments.
We adapted the article title to better reflect the scope as you recommended.
As we did not see those patients for follow-up examinations we can only report their uncomplicated full recovery that we confirmed by telephone follow-ups.
